# Asymmetry in kinematic generalization between visual and passive lead-in movements are consistent with a forward model in the sensorimotor system

Ian S. Howard[1]*, Sae Franklin[2], David W. Franklin[3]

**1** Centre for Robotics and Neural Systems, School of Computing, Electronics and Mathematics, University of Plymouth, Plymouth, England, United Kingdom, **2** Department of Electrical and Computer Engineering, Institute for Cognitive Systems, Technical University of Munich, Munich, Germany, **3** Neuromuscular Diagnostics, Department of Sport and Health Sciences, Technical University of Munich, Munich, Germany

* ian.howard@plymouth.ac.uk

**Data Availability Statement:** Data underlying the study is available from the Dryad repository (DOI: 10.5061/dryad.513bv1v).

## Abstract

In our daily life we often make complex actions comprised of linked movements, such as reaching for a cup of coffee and bringing it to our mouth to drink. Recent work has highlighted the role of such linked movements in the formation of independent motor memories, affecting the learning rate and ability to learn opposing force fields. In these studies, distinct prior movements (lead-in movements) allow adaptation of opposing dynamics on the following movement. Purely visual or purely passive lead-in movements exhibit different angular generalization functions of this motor memory as the lead-in movements are modified, suggesting different neural representations. However, we currently have no understanding of how different movement kinematics (distance, speed or duration) affect this recall process and the formation of independent motor memories. Here we investigate such kinematic generalization for both passive and visual lead-in movements to probe their individual characteristics. After participants adapted to opposing force fields using training lead-in movements, the lead-in kinematics were modified on random trials to test generalization. For both visual and passive modalities, recalled compensation was sensitive to lead-in duration and peak speed, falling off away from the training condition. However, little reduction in force was found with increasing lead-in distance. Interestingly, asymmetric transfer between lead-in movement modalities was also observed, with partial transfer from passive to visual, but very little vice versa. Overall these tuning effects were stronger for passive compared to visual lead-ins demonstrating the difference in these sensory inputs in regulating motor memories. Our results suggest these effects are a consequence of state estimation, with differences across modalities reflecting their different levels of sensory uncertainty arising as a consequence of dissimilar feedback delays.

**Funding:** Financial support for ISH was provided by the Centre for Robotics and Neural Systems at the University of Plymouth and by the Bavarian State Ministry for Science, Research & the Arts. The funders had no role in study design, data collection and analysis, decision to publish, or preparation of the manuscript.

**Competing interests:** The authors have declared that no competing interests exist.

## Introduction

Recent studies have highlighted key aspects for neural rehabilitation using robotic systems [1]. However, continual progress in this area depends on understanding the mechanisms of human sensorimotor learning in order to determine the optimal presentation of sensory information to improve the rate, retention and generalization of adaptation. Although adaptation is often studied on single movements in the laboratory, we rarely produce movements in isolation in everyday life. Rather, one movement often directly leads into another. For example, to catch a ball, we make use of visual motion information to estimate its state in order to plan and execute an interception movement. Thus, natural movements often follow directly from previous movements, or from visual motion.

This scenario can be investigated in the laboratory by using two-part movements. Here, the first part consists of a lead-in movement from a start location to an intermediate via point, which is followed closely in time by a second movement to a final target location. Recent work has shown that closely linking multiple movements together in time in this fashion reduces interference in learning opposing tasks [2–5]. In particular, distinct past movements act like a contextual cue, enabling adaptation to opposing viscous curl fields when the adaptation movements are preceded by unique lead-in motions, each associated with one of the dynamics [2]. This shows that motor learning and recall depends not only on the current state of the arm during a movement, but also on its preceding states. Interestingly, active, passive or visual lead-in movements were all equally effective at reducing interference. This indicates that sensory feedback relating to motion is sufficient to affect adaptation, even when no active movement is involved. The contextual effect of this prior movement disappears as the time between lead-in and adaptation movements exceeds about half a second, indicating that the representation of past state decays quickly over time. This suggests a strong link between the representation of state and the theory of neural population dynamics [6,7].

Dynamic adaptation to a single force field occurs locally; after training in a specific movement, the recall of predictive compensation decreases as the movement angle [8–11] or distance [12–14] deviates from the training condition. The Gaussian-like angular generalization observed in these studies has also be found for lead-in movements, with different lead-in modalities exhibiting different characteristics, both in terms of their absolute level of influence, but also in their sharpness of tuning. In particular both active [15] and passive lead-in movements [10] show narrower and deeper tuning than visual lead-in movements [11].

Interference studies have been widely adopted to investigate contextual effects on motor learning, and to examine if contextual cues can assist in the learning of opposing dynamics [14,16–21]. Such interference paradigms are more sensitive to generalization effects of contextual cues than single field paradigms, and have been used effectively to examine the angular generalization characteristics of lead-in movements [10,11]. Using these paradigms it has been possible to extract features of the neural basis functions underlying dynamical adaptation, allowing for the development of simple computational models [4]. However, we still lack basic information on the generalization features of lead-in movements for different kinematics such as duration or distance.

Here, we first characterize the generalization of passive and visual lead-in movements across different kinematics using an interference paradigm. In two separate experiments, we examine generalization across distance and duration (and the dependent variable of speed) of passive and visual lead-in movements. Second, in order to gain insight into any commonality between the neural resources employed in passive and visual lead-in movements, we also investigate how adaptation transfers between these two different lead-in modalities.

## Methods

### Experimental design

**Subjects.**   Sixteen human participants were randomly allocated to two experimental groups that each performed one experiment. Eight participants (7 female, aged 24.8 ± 5.0 years, mean ± sd) performed the passive lead-in experiment. Eight further participants (6 female; aged 27.4 ± 6.7 years) participated in the visual lead-in experiment. All participants were right handed according to the Edinburgh handedness questionnaire [22], and naïve to the aims of the study. All participants provided written informed consent to the protocol before participating in the experiment, which had been approved by the University of Cambridge Ethics Committee. The methods were carried out in accordance with the approved guidelines. Some of the data (visual lead-in experiment) was previously presented in a conference paper [23].

**Apparatus.**   Experiments were performed using a vBOT planar robotic manipulandum and its associated virtual reality system [24]. Handle position is measured using optical encoders sampled at 1000 Hz, and motors operating under torque control allow the application of end-point forces. A force transducer (Nano 25; ATI), mounted under the handle, measures the applied forces, and its output signals were low-pass filtered at 500 Hz using analogue 4th pole Bessel filters prior to digitization. To reduce body movement, participants were seated in a sturdy chair in front of the apparatus and firmly strapped against the backrest with a four-point seatbelt. During an experiment, participants grasped the robot handle in their right hand while their right forearm was supported by an air sled, constraining arm movement to the horizontal plane. Participants could not view their hand directly. Instead veridical visual feedback was used to overlay images of the starting location, via point, final target, (all 1.25 cm radius disks) and a hand cursor (0.5 cm radius red disk) using the virtual reality system. This ensured that the visual cursor appeared to the participant in the same plane and at the same location as their hand. Data was collected at 1000 Hz and logged to disk for offline analysis using Matlab (Matlab, The MathWorks Inc., Natick, MA, USA).

**Force fields.**   In the adaptation movement, participants performed reaching movements either in a null field condition, a velocity-dependent curl force field [14], or a mechanical channel [25]. The curl force field was implemented as:

$$\begin{bmatrix} F_x \\ F_y \end{bmatrix} = k \begin{bmatrix} 0 & -1 \\ 1 & 0 \end{bmatrix} \begin{bmatrix} \dot{x} \\ \dot{y} \end{bmatrix} \qquad (1)$$

where the field constant k was set to a value of ±16 Nm⁻¹s, and the sign determines the direction (CW or CCW) of the force-field. Each participant experienced both force field directions. The direction of the force field was always associated with a specific direction of a prior contextual movement. The relationship between the contextual movement direction and curl field direction (CW/CCW) was counterbalanced across participants.

Mechanical channel trials were implemented using a spring constant of 6,000 Nm⁻¹ and a damping constant of 30 Nm⁻¹s perpendicular to the direction of motion throughout the movement between the central location and the final target. Channel trials were only produced on the movements to the 0˚ target with corresponding lead-in movements starting at 135˚ or 225˚, and never presented on consecutive trials.

### Protocol

Two separate experiments were performed to examine the generalization of the learning associated with one contextual movement to other contextual movements with different kinematic

profiles (within the same modality), as well as the transfer of learning between passive and visual lead-in conditions (across the modalities).

After an initial pre-learning session in a null field, participants were exposed to the curl force fields (learning phase). Channel trials were used to examine adaptation to the novel dynamics, in which the lead-in movement duration, speed and distance were varied. In addition, the modality of the lead-in movement was occasionally changed to examine transfer. The trial parameters for both experiments are shown in Table 1 and the kinematics of the lead-in movements can be seen in Fig 1. On these trials, the lead-in movement was chosen from one of 15 different movements with distances ranging from 3 cm to 20 cm and durations ranging between 210 ms to 1400 ms.

**Trial organization.** All trials consisted of a two-part movement: An initial lead-in movement followed directly by an adaptation movement. The first part was a contextual lead-in movement from a starting location to a central via point. This contextual lead-in movement was 10 cm in length during all null and force field training conditions. The second part was an 18 cm adaptation movement to the final target. The participants only experienced a force field or channel trial during this adaptation part of the movement. Defining angular locations relative to the via point, in each experiment there were two target locations, at 0˚ and at 270˚ respectively. The 0˚ target location was associated with lead-in start locations at +135˚ and 225˚, and the 270˚ target location was associated with lead-in start locations at +45˚ and 135˚. For each target location, the associated start locations were indicative of the direction of the curl force field on the adaptation movement (clockwise or counter-clockwise). All in all, this resulted in four possible distinct two-part movements (combinations of lead-in and adaptation movement).

**Table 1. Durations, distances and peak velocities of the lead-in movements for the training and generalization conditions.** Passive and visual lead-in movements are represented by P and V respectively. The P/V Training condition used a passive lead-in for experiment 1 and with a visual lead-in for experiment 2. In both experiments, test trials with a passive lead-in and a visual lead-in were performed with the same kinematics as the training conditions. Similarly, a P/V channels represents channel conditions used with a passive lead-in for experiment 1 and with a visual lead-in for experiment 2. The Reverse V Channel was used with a visual lead-in in both experiments.

| Condition | Duration [ms] | Distance [cm] | Peak velocity |
|---|---|---|---|
| P/V Null/Training | 700 ms | 10 cm | 26.8 cms-1 |
| P Channel | 700 ms | 10 cm | 26.8 cms-1 |
| V Channel | 700 ms | 10 cm | 26.8 cms-1 |
| P/V Channel | 1400 ms | 20 cm | 26.8 cms-1 |
| P/V Channel | 1050 ms | 15 cm | 26.8 cms-1 |
| P/V Channel | 420 ms | 6 cm | 26.8 cms-1 |
| P/V Channel | 210 ms | 3 cm | 26.8 cms-1 |
| P/V Channel | 1400 ms | 10 cm | 13.4 cms-1 |
| P/V Channel | 1050 ms | 10 cm | 17.9 cms-1 |
| P/V Channel | 420 ms | 10 cm | 44.6 cms-1 |
| P/V Channel | 350 ms | 10 cm | 53.6 cms-1 |
| P/V Channel | 700 ms | 20 cm | 53.6 cms-1 |
| P/V Channel | 700 ms | 15 cm | 40.2 cms-1 |
| P/V Channel | 700 ms | 6 cm | 16.1 cms-1 |
| P/V Channel | 700 ms | 3 cm | 8.0 cms-1 |
| P/V Channel | 1050 ms | 20 cm | 35.7 cms-1 |
| P/V Channel | 350 ms | 3 cm | 16.1 cms-1 |
| Reverse V Channel | 700 ms | 10 cm | -26.8 cms-1 |



**Fig 1. Kinematics of lead-in movements used for testing generalization.** **A** Profiles of movement distance versus duration of lead-in probe conditions across all conditions. Thick black line indicates the training lead-in motion. Colors indicate specific conditions that are matched across duration (red), peak speed (green), or duration (blue). **B** Lead-in movement kinematics of peak speed as a function of duration.

## Passive lead-in experiment

**A** Contextual phase | Adaptation phase

Pulled

CW Field

**B**

CCW Field

Pulled

**C** Contextual phase | Test phase

Pulled

Channel

## Visual lead-in experiment

**D** Contextual phase | Adaptation phase

Visual movement

CW Field

**E**

CCW Field

**F** Contextual phase | Test phase

Channel

Visual movement

**Fig 2. Experimental design. A-C:** Passive lead-in generalization experiment. **A** Participants first experienced an initial passive lead-in motion from a starting position (grey circle, shown here at 225˚) to the central target (green circle) and then immediately made a second active movement to the target (yellow circle shown here at 0˚) on which a curl force field (blue arrows) could be applied. **B** An initial movement from a different starting target (shown here at 135˚) was associated with the opposite force field on the second movement. The direction of curl force field and lead-in movements were counterbalanced across participants. **C** In order to examine learning rate and generalization, random trials in which the contextual movement (shown here at 225˚) was followed by a mechanical channel on the second movement to the target were applied. **D-F:** Visual lead-in generalization experiment. **D** Participants initially observed an initial visual cursor movement (red circle) from the grey starting circle to the central target (green circle). Once the cursor entered the central target, participants immediately performed a second active movement to the target (yellow circle) on which a curl force field (blue arrows) could be applied. **E** An initial cursor movement from a different starting target was associated with the opposite force field on the second movement. **F** On random trials, after the visual lead-in motion, a mechanical channel was applied on the active movement to the target to measure predictive compensation.

Fig 2A–2C shows start and target locations only for the 0˚ target location case. When the lead-in starts from the 225˚ location, the adaptation movement is associated with a CW field (Fig 2A). When the lead-in starts from the 135˚ location, the adaptation movement is associated with a CCW field (Fig 2B). Fig 2C shows one of the two possible channel trials, in this case with the lead-in starting from the 225˚ location (the other case is with a lead-in from the

135˚ start location but is not shown here). This relationship between starting location and the CW and CCW field directions shown here was switched for half of the participants, to counterbalance the relationship of lead-in direction and field direction across the experiments.

**Session design.**   As training an interference paradigm can take a significant number of trials and a large number of channel trials were required to collect the generalization data and enable low variance estimation of compensation to be made, each experiment was performed in two separate sessions on different days. This procedure was adopted to limit fatigue effects that might have been experienced from one single long experimental session. As some forgetting may result between sessions, the second session commenced with another training phase before proceeding with the generalization phase. In total, there were 1546 and 1580 trials on days 1 and 2 respectively, leading to a total of 3126 trials overall. In particular there were 2248 training field exposure trials and 544 channel trials use to probe generalization characteristics.

**Probe trials.**   In order to probe the generalization characteristics there were 17 different lead-in conditions for the channel trials (Table 1). Each lead-in was performed from the two possible starting locations to probe recall of both the CW and CCW field contexts. This gave rise to a total number of 34 distinct channel trial lead-ins (17 x 2 starting locations). Each of these 34 distinct channel trials was repeated 16 times during the generalization phases of the experiments. These generalization lead-in conditions were always followed by a channel trial on the adaptation movement. Fifteen of these seventeen lead-in movements were chosen to sample lead-in distances between 3–20 cm, peak speeds between 8.04–53.57 cms$^{-1}$ and durations between 210–1400 ms (Fig 1). One of these 15, had the exact motion as the training movements (Fig 1, black line). The sixteenth lead-in movement was designed to test transfer between passive and visual lead-in movements. That is, in the passive lead-in experiment, this lead-in was a visual lead-in with the same kinematics as the training movement. In the visual lead-in experiment this was a passive lead-in movement. Finally, the seventeenth condition was a reversed visual cursor condition. This "lead-away" condition was similar to a lead-in movement between a start and via-position, except the cursor started at the central via-point and moved to what was previously the start position. This condition examined if the reversal of visual motion would still lead to a recall of dynamics during the second probe phase, or whether the form of visual movement needed to be consistent with the lead-in movement used during training to elicit an appropriate contextual effect or transfer. Both the trained lead-in probe trial and the opposite modality lead-in probe trials were also repeated throughout the pre-exposure and training phase of experiment, such that each was repeated a total of 38 times. The trained lead-in probe trials provided a means to assess learned compensation as the experiment progressed.

**Trial block organization.**   The experiment was organized in blocks. These blocks contained different numbers and types of trials in the pre-exposure, training and test phases of the experiment. Within a block, trials were sorted pseudo-randomly, with the constraint that channel trials were not allowed to be adjacent or the first trial in a block. Participants were required to take short rest breaks approximately every 200 trials (195–205 trials) but could rest at any time between trials. The blocks and trials were organized as follows:

## Day 1

**Pre-exposure.**   The pre-exposure phase started with 2 blocks of 40 trials. A block consisted of 36 Null trials and 4 channel trials. The four channel trials were 2 channel trials in the training condition (one for each lead-in direction) and 2 channel trials in the transfer

condition (that is, in the other modality, with one trial for each lead-in direction). The 36 null trials were evenly split between movements to the 0˚ direction target and the 270˚ direction target, with equal numbers of each lead-in direction for each target (9 repetitions of each of the four trial types). Next, participants were provided with three repetitions of each of the 17 generalization channel trials for each of the two possible lead-in directions (102 generalization condition channel trials) to ensure prior experience of all the generalization trial conditions. Finally, 2 blocks of 40 trials were again performed as described above (36 null and 4 channel).

**Field-exposure training.** During the exposure phase, participants were exposed to the curl force fields during the adaptation movement. This phase consisted of 12 blocks of 40 trials (36 field trials and 4 channel trials in a block) arranged as in the pre-exposure phase. This was a total of 480 trials (432 force field trials, 24 training condition channel trials and 24 transfer condition channel trials).

**Generalization testing.** This phase examined generalization of the learned predictive compensation by pseudo-randomly interspersing curl force field trials with trials in which channel trials were preceded by the full range of contextual 34 movements (17 different generalization trial types x 2 lead-in directions). The generalization phase consisted of 6 blocks of 134 trials. Each block consisted of one of each of the 34 channel trials and 100 curl force field trials (25 of each of the four types). This provided 6 repetitions of each of the 34 generalization channel trials. In this phase there was a total of 804 trials (600 curl field trials and 204 generalization channel trials).

### Day 2

**Exposure training.** At the beginning of the second session, training was briefly resumed. The phase consisted of 6 blocks of 40 trial blocks (36 curl field trials + 4 channel trials) for a total of 240 trials.

**Generalization testing.** Similar to the session of Day 1, participants performed 10 blocks of 134 trials (Total 1340 trials: 1000 curl field trials and 340 generalization condition channel trials), such that each of the 34 different probe trials (17 x 2 lead-in directions) was repeated 10 times.

### Experiment 1. Passive lead-in movements

In experiment 1, the contextual lead-in movement was comprised of passive movement of the participant's hand. This passive movement was produced by the robotic manipulandum passively moving the participant's hand while no cursor was presented. Each trial began by displaying the start location for the lead-in movement, the central location and final target. The vBOT then moved the participant's hand to the lead-in start location. Once the handle was stationary within the start location for 300 ms, a beep was generated indicating the start of the trial. At this time, the handle of the robotic system moved to the central via-point following a minimum jerk trajectory. The training contextual movement was a 10 cm movement of duration 700 ms. Once the hand reached the central location, participants were required to produce an active adaptation movement from the central location to the final target location. The dwell time of the hand within the central via point was required to be between 0–250 ms, otherwise a warning was provided. If dwell time exceeded 500 ms then the trial was aborted and repeated. If the second movement (adaptation movement) duration was between 450 ms and 600 ms a "Great" message was displayed; otherwise an appropriate "Too Fast" or "Too Slow" warning was shown. Force fields and channel trials were only ever presented during this second movement.

## Experiment 2. Visual lead-in movements

Experiment 2 had a similar design to Experiment 1 and used the same block structure. The only difference was that the contextual lead-in movements for both training and generalization testing consisted of a visual movement of the cursor. This is illustrated in Fig 2D–2F for the 0˚ target location condition. The training contextual lead-in movement again followed a minimum jerk trajectory of duration 700 ms from the start to the central location. During this time, the participant's hand remained stationary at the central location. Immediately after the cursor reached the central location, the participant made an active reaching adaptation movement from the central location to the final target. The same variations of generalization movement trials were performed (but with visual instead of passive motion). In addition, a transfer condition was used in which a passive movement lead-in was performed. Again, a reversed visual cursor condition was also employed.

## Participant instructions

Verbal instructions were used to instruct participants how to take part in the experiments. Participants were first asked to read the ethics form and sign if they wished to proceed with the experiment, which would last a few hours over two days.

Participants in the passive condition were informed that at the start of a trial that the robot would pull them to the via point and they were required to immediately try to move to the target. They were told that this robot pulled movement could come from different directions or with different lengths of movements, but that their task was the same–to make a movement from this via-point to the target. They were asked to move briskly and that feedback on movement speed and any start delay would be reported after each trial. Specifically, they were informed that the movement should be a point-to-point goal directed reaching movement completed in a single movement without overshooting or undershooting the target. They were informed that they should go roughly straight to the target using a natural relaxed movement. They were also told that at some point during the experiment some kind of disturbance would occur in which the machine might push them to the left or the right of their movement and disturb their trajectory. They were also informed that to complete the movement they need to get the cursor to the target regardless of this disturbance. Finally, they were also informed that approximately every 200 trials there would be a short break. They were also made aware that they could take a break at any time if they released the handle switch on the robot. Instructions for the visual condition were identical, except participants were informed that on commencement of a trial the cursor would move to the center position while their hand was stationary at a via point, and then they were required to try to move quickly to the target, as in the passive experiment.

## Data analysis

The experimental data was analyzed offline using Matlab R14. Statistics to examine differences between the generalization from visual lead-in and passive lead-in movements were performed in JASP 0.11.1 (JASP Team, 2019) using a repeated measure ANOVA. T-tests were performed within Matlab. To examine learning, kinematic error on the adaptation movements and force compensation on the channel trials were used.

**Kinematic error.** The kinematic error was calculated on the adaptation portion of the movement, but only during the null and curl field trials. To be consistent with our previous work, this was quantified as the maximum perpendicular error (MPE) for each trial, which is the maximum deviation of the hand path to the straight line joining the movement starting location to the target. It was necessary to have different experimental block sizes in various

phases of the experiment to accommodate the training and generalization trials. Therefore, to calculate a consistent averaged-across-trials of MPE, an analysis block size of 8 was chosen, because this number of trials could be used consistently throughout the experiment. During calculation of the analysis block MPE average, the sign of each trial MPE was flipped appropriately so that results from CW and CCW field trials could be averaged together. The mean and standard error (SE) of the averaged-across-trials MPE values was then computed across all participants.

**Force compensation.** On each channel trial, during the adaptation movement between the via point to the final target, velocity of the movement towards the target and the force exerted by participants perpendicularly into the wall of the simulated channel were simultaneously recorded at 1000Hz, to enable the estimation of predictive feedforward adaptation [25], which is an established technique in the analysis of dynamic learning in viscous curl fields [26]. Using this paradigm, the channel clamps movement perpendicular to the direction of movement to small values by resisting it using a high spring constant (6000 Nm$^{-1}$) as well as viscous damping (30 Nm$^{-1}$s), thereby minimizing lateral movement error. As there is no lateral error on these trials, there is no error-induced feedback component. Since channel trials occur sparsely within blocks of curl fields trials, this means that any measured perpendicular force exerted into the channel will be dominated by the predictive feedforward force that has been learned to compensate the curl field.

To estimate the level of force field compensation of the participant, the measured perpendicular channel force samples during movement along the channel were regressed against their corresponding forward velocity samples scaled by curl field strength. This calculation was performed for each movement over the period from leaving the via-point until entering the target. This yielded an estimate of the level of force compensation present in each channel trial [26]. For each participant, the force compensation values were averaged across 2 sign-corrected sequential channel trials (since they corresponded to opposite curl field directions). The mean and standard error (SE) of compensation was then computed across participants. This method to assess adaptation to the novel dynamics is preferable to relying on a reduction in kinematic error during force field learning, since the latter can also arise from muscle co-contraction [27–29].

## Results

In the passive lead-in experiment, participants performed active reaching movements to a target after being passively moved from a start position to a central target. After initial movements in a null field, participants were presented with a curl force field during the active movement. The direction of the curl field depended on the angle between the passive movement and active movement (Fig 2A and 2B). When presented with the curl force field, participants' adaptation movements were disturbed, producing large errors that were gradually reduced over the exposure phase (Fig 3A). Throughout the experiment channel trials were introduced on random trials in order to measure the predictive force compensation throughout adaptation (Fig 2C). Over a similar timescale as the reduction in kinematic error, force compensation increased, reaching just over 62% compensation averaged over both force fields (Fig 3B). A small but significant increase in the kinematic error can be seen between day 1 and the start of day 2 (paired t-test between final block on day 1 and first block on day 2: $t_7 = 3.49$; $p = 0.01$), but the associated decrease in force compensation was not significant ($t_7 = 2.09$; $p = 0.075$). Compensation to each field direction separately is shown in Fig 3F. The final levels of force compensation (mean of final 4 blocks) were not significantly different between the two lead-in directions (paired t-test: $t_7 = 0.743$; $p = 0.48$).

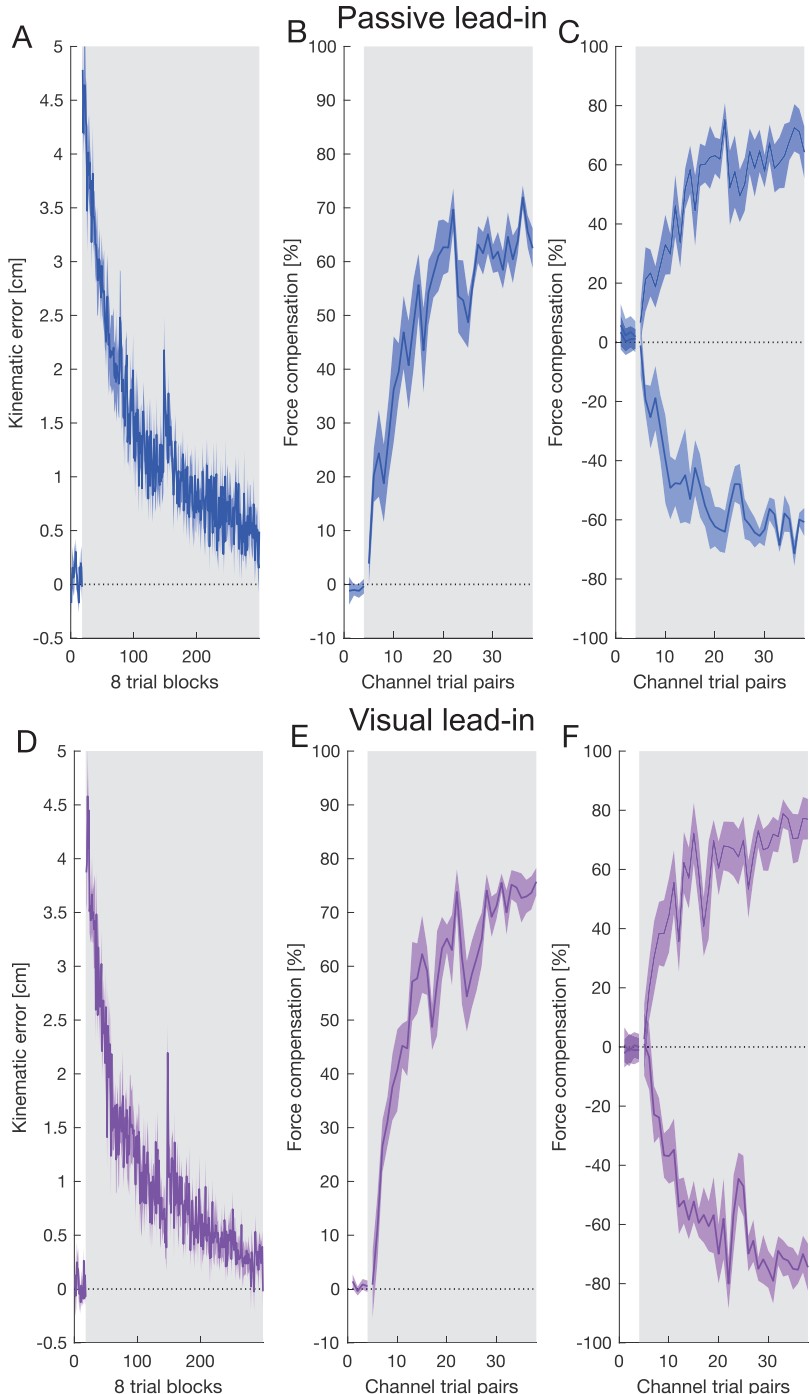

**Fig 3. Adaptation to two opposing force fields. A** Mean and SE of MPE across over 8 participants for the passive lead-in experiment as a function of blocks of 8 trials. **B** Mean and SE of percentage force compensation for pairs of channel trials (one for each force field direction) throughout the passive lead-in experiment where the lead-in movement was the same as the training trials. **C** Mean and SE of percentage force compensation as B but showing compensation for each field direction separately. **D** Mean and SE of MPE for the visual lead-in experiment. **E** Mean and SE of percentage force compensation for the visual lead-in experiment. **F** Mean and SE of percentage force compensation as E but showing compensation for each field direction separately.

Participants in the visual lead-in experiment performed a similar protocol but where the lead-in movements were purely visual in nature (Fig 2D–2F). Again, when presented with the curl force field, participants' adaptation movements were disturbed, producing large errors that were gradually reduced over the exposure phase (Fig 3D). Again, a small but not significant reduction in the force compensation ($t_7$ = 1.67; p = 0.14) and a small but significant increase in the kinematic error ($t_7$ = 4.19; p = 0.004) can be seen between day 1 and the start of day 2 (final block day one compared to first block day 2 with paired t-test). Over a similar timescale, force compensation increased, reaching approximately 70% compensation averaged over both force fields. Mean and SE of percentage force compensation showing compensation for each field direction separately is shown in Fig 3F. The final levels of force compensation (mean of final 4 blocks) were again not significantly different between the two lead-in directions (paired t-test: $t_7$ = 0.402; p = 0.70).

On random trials late in the adaptation phase, channel trials were applied with a range of different lead-in movement kinematics (Fig 1) in order to examine generalization. After learning the force fields with the passive lead-in movement, variations in the kinematics of this lead-in movement produced a range of generalization levels (Fig 4A). As the testing lead-in movements varied further away from the training kinematics the predictive force level decreased. A repeated measures ANOVA with a within subject effect of lead-in kinematics (15 levels) demonstrated a significant main effect ($F_{14,98}$ = 33.057; p<0.001). A similar finding is shown for the generalization after learning a visual lead-in movement (Fig 4B). Although in this condition only small decreases in the predictive force are seen over a wide range of changes in the lead-in kinematics, the repeated measures ANOVA again showed a significant main effect of lead-in kinematics ($F_{14,98}$ = 23.397; p<0.001) demonstrating that the lead-in kinematics affected the predictive force on these channel trials.

Across the different lead-in movement kinematics, several conditions had the same duration, peak velocity or distance as the learned training condition (dotted lines in Fig 4). We examined the predictive force compensation values over these conditions in more detail along iso-contours for lead-in distance, duration and speed (Fig 5). Significant differences for select comparisons are reported from post-hoc tests following the significant main within subject effect of the repeated measures ANOVA reported above (Holm corrected for multiple comparisons across all 15 levels). The results for the passive lead-in condition show strong variations over changes in lead-in duration, distance and speed (Fig 5A–5C).

In panel A, lead-in distance is constant and the movement duration (shown on the x-axis) and speed vary. In panel B, lead-in duration is constant and the movement distance (shown on the x-axis) and speed vary. In panel C, lead-in speed is constant and the movement duration (shown on the x-axis) and the distance vary. The dotted lines indicate the training values of lead-in distance and lead-in duration. It can be seen that the recall of predictive compensation was strongly affected by changes in the duration of the movement (shown for constant distance conditions in Fig 5A and constant speed conditions on Fig 5C). There was a strong tuning effect centered around the movement duration used for training (Fig 5A). Specifically, compared to the training value, the force compensation significantly decreased as movement duration either increased to 1.4s (p = 0.016) or decreased to 0.35s (p = 0.033). Similar results were seen for a constant peak speed (Fig 5C), but while compensation significantly decreased as movement duration decreased to 0.21s (p = 0.01) this did not reach significance for the longer duration (p = 0.062). However, changing lead-in distance produced different effects. While reducing lead-in distance to 3cm reduced compensation (p = 0.004), increasing movement distance from the training value to 20cm had essentially no effect (p = 1.0) as shown in Fig 5B.

In the visual lead-in condition, we found a slightly less pronounced reduction in recalled compensation as the some of the kinematics were varied while others remained constant

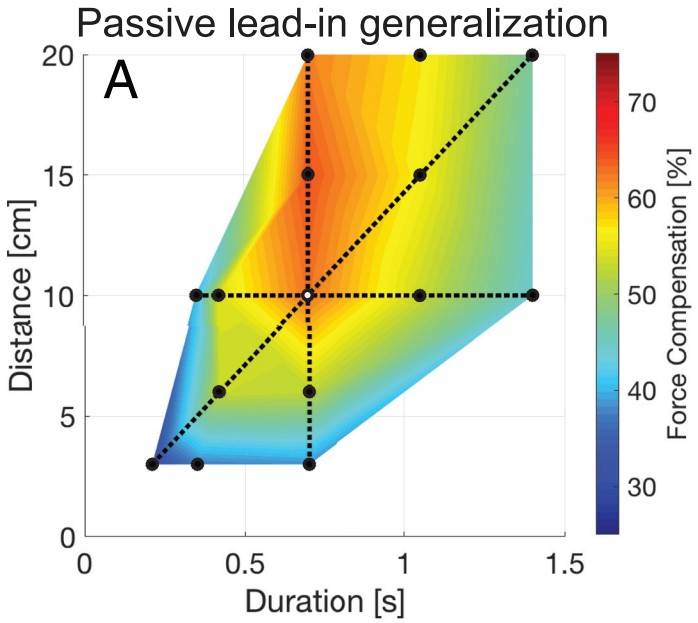

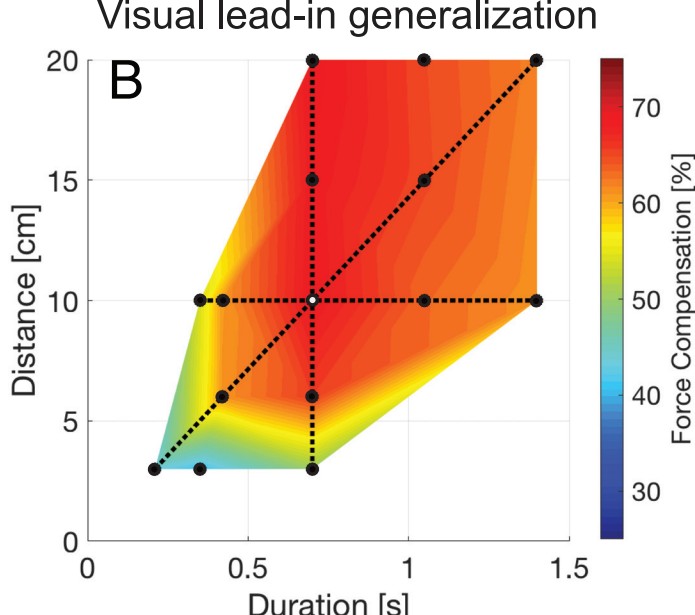

**Fig 4. Generalization surface plots for passive and visual lead-in movements. A** Surface plot of generalization for passive lead-in movements. The percentage force compensation is represented by color and plotted against lead-in duration and lead-in distance. The black circle with a white center indicates the result at the training condition. The solid black dots indicate points for which measurements were made on probe trials. The black dotted lines correspond to conditions with the same training lead-in distance of 10 cm, same training lead-in duration of 0.7s or same training lead-in speed of 26.8 cm/s. The legend shows the correspondence between color and percentage perfect force compensation. **B** Surface plot of generalization for the visual lead-in condition.

(Fig 5D–5F). Again, it can be seen that compensation was affected by changes in the duration of the movement, with compensation significantly falling off as movement duration decreased to 0.35s (p<0.001). Although there was a tuning effect centered around the movement duration used for training, as duration deviated from the training value the fall off was appeared

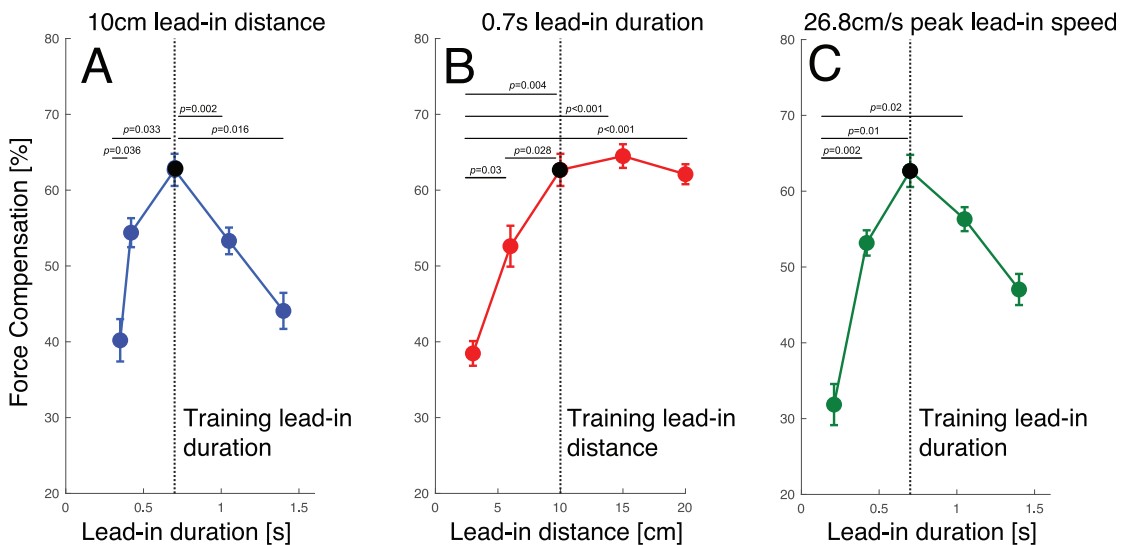

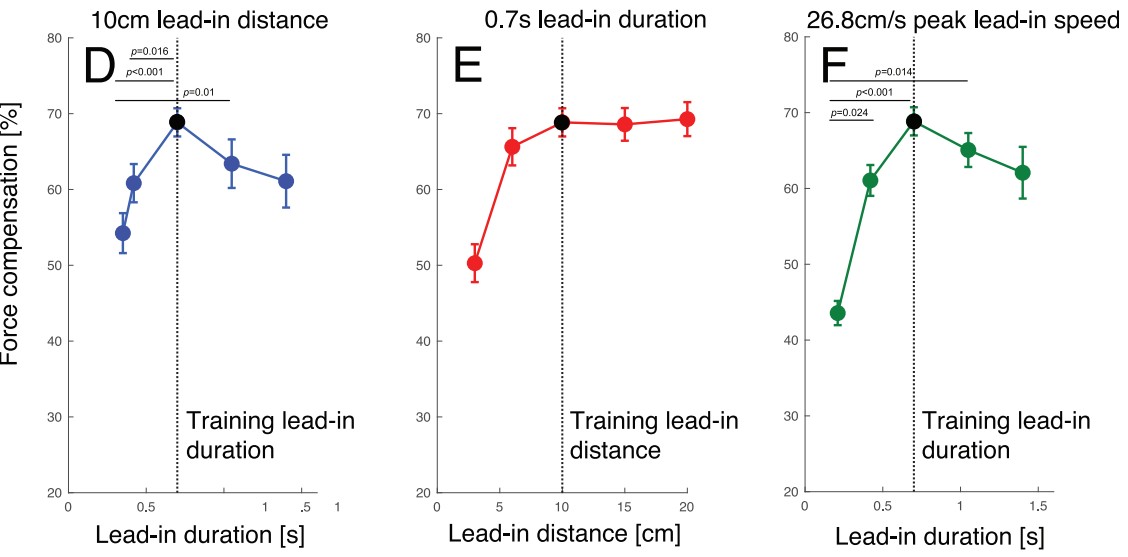

**Fig 5. Generalizations results for both passive and visual lead-in conditions plotted for fixed values of lead-in distance, lead-in duration and lead-in speed. A-C** Results of passive lead-in experiments. The dotted lines indicate the training values of lead-in distance and lead-in duration. Error bars indicate standard error of the mean. Significant differences are indicated for post-hoc comparisons (Holm corrected for multiple comparisons). **A** Effect of changing lead-in duration for fixed 10 cm lead-in distance. In this panel, lead-in distance is held constant and the movement duration (shown on the x-axis) and speed vary. **B** Effect of changing lead-in distance (and peak speed) across conditions with a fixed 700 ms lead-in duration. **C** Effect of changing lead-in duration (and distance) across conditions with a fixed 26.8 cm/s lead-in speed. **D-F** Corresponding results for visual lead-in condition.

less pronounced than in the passive lead-in condition (Fig 5D). Similar results were seen for a constant peak speed (Fig 5F), with compensation significantly falling off as movement duration decreased to 0.21s (p<0.001). However, as the training distance varied but duration was fixed (Fig 5E, we found no significant differences (all p>0.1).

After adaptation in the two experiments, the asymptote of force compensation was slightly higher in the visual lead-in (68.8 ± 5.3%) compared to the passive lead-in (62.7 ± 6.0%) conditions (Students t-test, $t_7$ = 2.191; p = 0.046). If we compare the generalization of the predictive compensation across the two experiments, we can see one major finding; namely that the overall tuning effects were more pronounced for the passive lead-in condition compared to the visual lead-in condition. Visual lead-in generalization showed less sensitivity to variations in lead-in kinematics. Indeed, whereas passive lead-ins resulted in a 2D monotonic curved surface in the dimensions of duration and distance (Fig 4A), the corresponding surface for visual lead-ins (Fig 4B) exhibits a large region consisting of a flat planar surface. To test if this difference in the generalization across kinematics between passive and visual lead-in movements is statistically significant, we performed ANOVAs on the force compensation results (with main effects of kinematic condition (14 levels) and lead-in modality (2 levels: visual or passive). To do so, the force compensation for each testing condition was normalized with respect to the value at the trained condition for each participant. We then compared the generalization across kinematic conditions using a repeated measures ANOVA with a repeated measure factor of lead-in condition (15 levels) and between subject factors of experimental condition (2 levels: passive and visual lead-ins). We report the Greenhouse-Geisser sphericity corrected values. There was both a significant main effect of lead-in condition ($F_{5.127,71.782}$ = 56.627; p<0.001), experimental condition ($F_{1,14}$ = 8.073; p = 0.013), and interaction between these two ($F_{5.127,71.782}$ = 3.061; p = 0.014). As the predicted force compensation level was normalized to 100% for the training condition, the presence of a significant main effect of experimental condition demonstrates that there were differences between the shape of this generalization between the passive and visual lead-in movements. This highlights a clear difference between visual inputs and passive inputs as a contextual signal for motor adaptation, extending our previous findings [10,11].

We also investigated how learning opposing force fields with contextual cues in one sensory modality would transfer to the other sensory modality. To investigate this, occasional channel trials were used with a lead-in in the other modality. It can be seen that there is asymmetric transfer between passive lead-in and visual lead-in movements (Fig 6). Although there was partial transfer from passive to visual lead-in movements (Fig 6A) with values reaching just above 20%, there was much less transfer from visual to passive lead-in movements (Fig 6B) with values just under 10%. To compare the level of transfer between the two modalities, the transferred adaptation was scaled according to the final level of adaptation in each experiment and compared using a t-test. The transfer from passive to visual (35.2% ± 12.1%; mean ± std) was significantly larger ($t_{14}$ = 3.966; p = 0.0014) than the transfer from visual to passive (13.0% ± 10.2%). Thus, there is a clear asymmetry between the transfer of adaptation between these two sensory modalities.

Finally, we examined how learning the visual lead-in movement would transfer to a completely reversed visual cursor (with the same duration and distance). To balance conditions and the number of trials across experiments, this was also tested for the passive lead-ins. To compare between the two experiments, the predictive compensation of the reversed visual cursor was scaled by the predictive compensation on the training condition. There was no significant difference in transfer to reverse visual movement between participants trained with passive lead-in movements (22.7% ± 17.3; mean ± std) versus those trained with visual lead-in movements (29.3% ± 23.0) using a t-test ($t_{14}$ = 0.641; p = 0.53). This shows that the predictive compensation is sensitive to the direction of the visual motion.

## Discussion

We investigated the kinematic generalization characteristics of passive and visual lead-in movements using a force field interference paradigm. Participants first experienced a lead-in

 Spatiotemporal generalization of lead-in movements

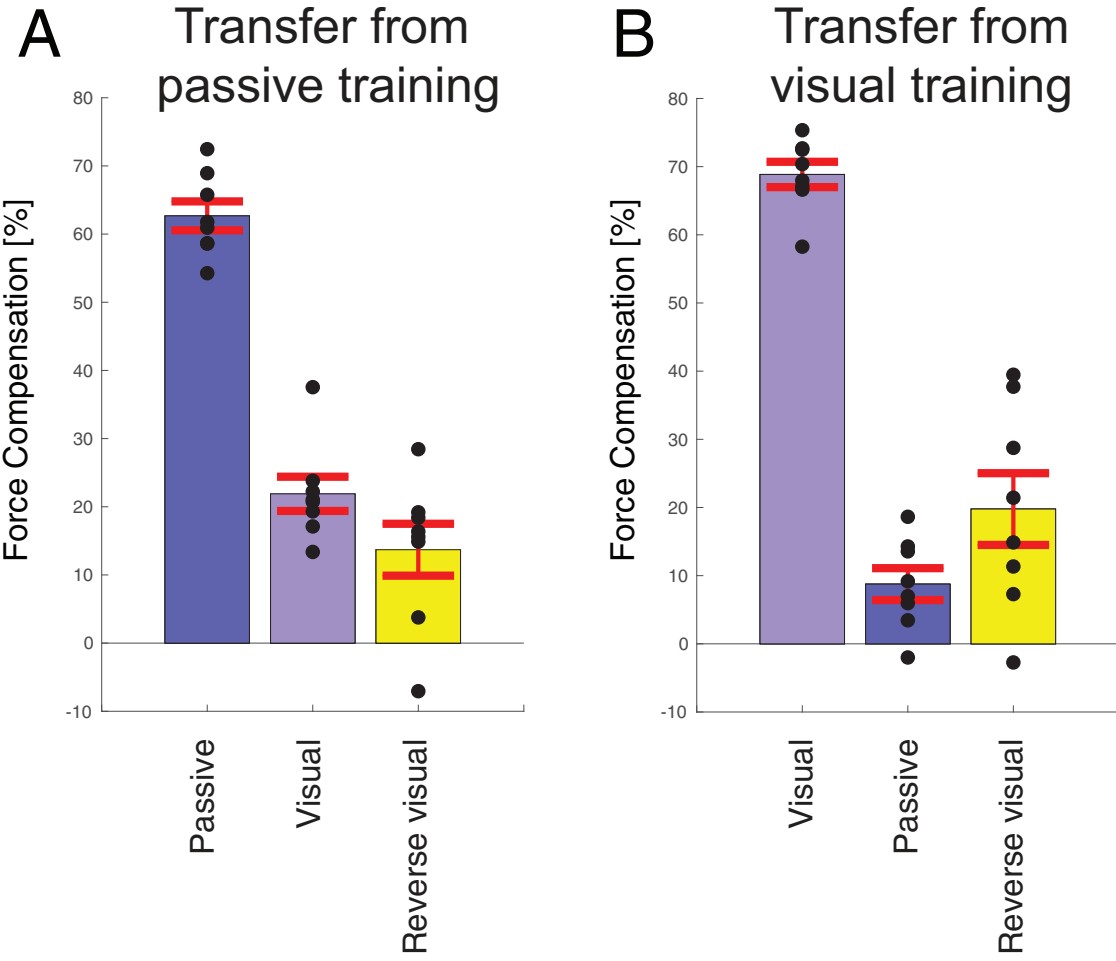

**Fig 6. Comparison of transfer across sensory modality or to reversed visual cursor. A** Transfer from passive lead-in to visual lead-in or reversed visual lead-in. For comparison the learned force compensation on passive movements is shown. Black circles indicate the results of individual participants. Error bars indicate standard error of the mean. **B** Transfer from visual lead-in to passive lead-in or reversed visual lead-in.

movement and then immediately made an active movement in a curl force field where the field direction was associated with the lead-in movement. Channel trials within the active movement examined how predictive compensation varied as lead-in kinematics were varied. In the first experiment lead-in movements were passive, whereas in the second experiment they were visual. For both modalities, recall of predictive compensation decreased as the duration of the lead-in movements deviated from the training condition. Reducing lead-in distance also reduced compensation but increasing lead-in distance had little effect on the force generalization. Our results show that although passive and visual lead-in movements influence memory formation and recall in subsequent movement, passive motion exhibits narrower generalization characteristics, whereas visual motion is less sensitive to kinematic change.

These generalization results further characterize the neural tuning exhibited by lead-in movements, extending beyond the directional tuning seen previously. The observation that passive lead-ins were more sensitive to changes in kinematics than visual lead-ins is consistent with the prior observations examining angular generalization [10,11,15]. Namely that active and passive tuning was narrower than the wider tuning seen in the visual condition.

PLOS ONE | https://doi.org/10.1371/journal.pone.0228083   January 29, 2020   16 / 21

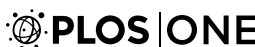

Here we describe this gradual change in the predictive force output on the adaptation movement as the lead-in motion is changed as generalization. We suggest that this generalization arises due to implicit learning mechanisms within the sensorimotor system. Although it has been shown that explicit strategies can play a role in learning visuomotor rotations [30–32], studies in force field learning have shown very little contribution of these explicit strategies [33]. Could the participants somehow predict the presence of channel trials and therefore stop to compensate for the force field? As the participants have no knowledge of the experiment, including the existence of channel trials interspersed between the field trials, and never experience errors during these specific trials, we suggest that this would be extremely unlikely. Moreover, we find no decrease in the force compensation between the very first block of generalization trials to the full results over the two-day experiment. Nevertheless, it is useful to consider alternative explanations to our observations. For example, while the direction of the force field depended on the lead-in direction, variations in the lead-in motion were never used to produce changes in the forces arising from the force field. Thus, one simple cognitive strategy could be that participants have simply learned a binary left-right mapping to the force fields, in which case we would see a constant predictive compensation for all variations of the lead-in kinematics. However, we discount this binary association hypothesis since it can be seen that variations in the kinematics of the passive (and to a lesser extent visual) lead-in movements produced clear variations in the predictive forces. This shows that the learning of this mapping was at least partially local and associated with the specific kinematics of the lead-in movement. The continuous nature of this effect, which exhibits a smooth transition in compensation between training conditions, also agrees with what was seen previously for angular generalization for both passive [10] and visual lead-in movements [11].

More recently it was shown that the tuning characteristics of different lead-in modalities could explain why angular variability of active lead-in movements affects the learning rate in two-part movement tasks, whereas no such effect exists for visual lead-in movements [4]. Our current results suggest that variations in the speed or duration of lead-in movements could provide similar decrements in learning rate, whereas an increase in movement distance would not have much effect. This might have important implications for rehabilitation, suggesting learning and recovery would be faster for training routines with consistent lead-in kinematics. One caveat is that such routines might also produce less generalization across tasks, as the adaptation is more likely to be learned specifically for the trained lead-in movement.

In both experiments, to examine transfer of adaptation across modality, a visual lead-in cursor motion occasionally replaced the passive lead-in, and vice versa. Interestingly, there was an asymmetric transfer between passive and visual lead-in movements, with partial transfer from passive lead-in movements to visual lead-in movements, but almost no transfer from visual to passive lead-in movements. Transfer could arise because passive lead-in movement partially engages neural mechanisms shared by the visual observation of movement, but not the converse. This result may be due to asymmetry in the connections between the neural substrates. Alternatively, it could arise because the visual feedback pathway has a lower gain due to the uncertainly introduced by the longer time delay associated with visual information [34]. The current observation that passive lead-in are more strongly tuned in duration than visual lead-ins, as well as the former results that the absolute level of influence of passive lead-ins [10] is higher than for visual lead-in [11] supports the latter hypothesis.

The wide ranging results from studies examining contextual cues for learning opposing dynamics have demonstrated that not all sensory signals are able to influence motor learning [15,20,35–41]. For example, color has essentially no effect [20]. In addition to the strong effects of prior movements, it has been shown that particularly effective contextual cues relate to state; for example limb state and physical locations [42,43], or different visual locations of the

cursor and targets [37]. Indeed, a location cue could constitute a complete physical shift of the movement task, or just a shift of one of its two essential components; namely a change in the location of the visual feedback, or a change in the physical location of the task with identical visual feedback. Other experiments have shown that future state also effects motor learning in an analogous way [3], with this effect depending on movement planning rather than execution [5].

On the face of it, it appears that there are multiple types of contextual cues that strongly influence motor memory formation. Here we propose that a factor they all have in common is that they are related to either past, current, or future state of the limb; or are signals used in the estimation of such limb states. That is, setting up the sensorimotor system in a different state before (or at the end of) a movement allows the formation and recall of different motor memories. This suggests that some contextual cues (such as visual lead-in movement or location in the visual workspace) are simply effective because the motor system makes use of these signals within a state estimation framework to determine the state of the arm. Such state estimation can only be made on the basis of sensory feedback and efference copy. This hypothesis would be consistent with the observation that visual or proprioceptive movements are as effective an active movement. It would also explain why a visual change of state can be as effective as a complete change in the physical state of the limb. Moreover, it can explain why vestibular inputs could also be used to learn opposing dynamics [15] but why color cues have much less effect on the adaptation system [20].

In order to reach with our arm to a specific location, our sensorimotor control system needs to know the initial limb state, and then activate the appropriate muscles in a specific pattern to generate forces that bring the arm into the final state to meet the task requirements. To make this movement robust in the face of noise and disturbances, this process does not simply rely on feedforward control, but makes use of sensory feedback of our arm's state, enabling online correction in any task-relevant deviation from the goal of our movement. Arm state can be estimated through the combination of appropriate sensory feedback signals such as proprioception from the skin muscles and joints, visual information, and vestibular inputs. However due to neural signal transmission and processing delays, motor responses to proprioceptive and visual feedback only start producing force after delays of 50 ms and 140 ms respectively. Such delays represent a challenge in the design of a feedback control systems, since using direct feedback from delayed signals can lead to instability.

To deal with delays in sensory feedback, Smith proposed an architecture which involves using immediate feedback from the output of a forward model of the plant, rather than the actual output of the plant itself [44]. In this architecture, the forward model estimates plant output without delay, thereby avoiding the instabilities that delay can introduce, Miall and Wolpert suggested that the Smith predictor architecture could account for delays in the human motor system [45,46]. In control engineering, forward models form part of observers, which are systems used to estimate the full state of a plant, which often cannot be observed directly. Such full state estimation can be used as the basis of full state feedback control of a plant, which is more effective than only basing control on the observable plant output. As with the Smith's predictor, such observers can also be constructed to estimate plant state without the delay, providing an elegant way to control plants that have inherent delay in their sensory feedback paths.

To make an observer robust to inaccuracies in its forward model and to deal with unpredictable disturbances, there is normally a state correction pathway term based on actual output error calculated as the difference between the actual output of the plant, and an appropriately delayed prediction of plant output. This results in the state prediction based on the efference copy of the motor command being combined with a state prediction error correction term

based on the delayed sensory feedback, which is something that has been shown to occur during the control of human movement [47].

Within such an observer-based controller framework, the observer performs state estimation for an active movement using efference copy, while improving the estimate using the delayed feedback signals. In the case of a purely visual observation or passive movement of the arm, the observer can still make a state estimate, but only based on the state corrections from feedback. From the premise that state is the key issue in formation of separate motor memory, such a framework would account for the observation that either active, passive or visual lead-in movements would influence state estimation.

To conclude, we have shown the current and previous observations of lead-in phenomena are consistent with the hypothesis that the human motor system operates as an observer-based controller mechanism, that makes use of a forward model to estimate state. In particular, our results support the proposal [34] that even though the variances of visual positional information is known to be lower than that obtained from proprioception, its longer temporal delay reduces its weighting in state estimation. As a consequence of this, visual information has less effect on the motor system than proprioceptive information, an effect that we have extended to the learning and generalization of opposing dynamics.

## Author Contributions

**Conceptualization:** Ian S. Howard, David W. Franklin.

**Data curation:** Ian S. Howard, Sae Franklin.

**Formal analysis:** Ian S. Howard, David W. Franklin.

**Funding acquisition:** Ian S. Howard, Sae Franklin, David W. Franklin.

**Investigation:** Ian S. Howard, Sae Franklin, David W. Franklin.

**Methodology:** Ian S. Howard, David W. Franklin.

**Project administration:** Ian S. Howard, Sae Franklin.

**Resources:** Ian S. Howard, David W. Franklin.

**Visualization:** Ian S. Howard, David W. Franklin.

**Writing – original draft:** Ian S. Howard, Sae Franklin, David W. Franklin.

**Writing – review & editing:** Ian S. Howard, Sae Franklin, David W. Franklin.

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
