## [Decision Letter · Decision Letter 0]

20 Sep 2019

PONE-D-19-15362

Asymmetry in kinematic generalization between visual and passive lead-in movements are consistent with a forward model in the sensorimotor system

PLOS ONE

Dear Dr. Howard,

Thank you for submitting your manuscript to PLOS ONE. After careful consideration, we feel that it has merit but does not fully meet PLOS ONE’s publication criteria as it currently stands.

Both reviewers have serious concerns, so please bear this in mind when preparing your revisions.

We would appreciate receiving your revised manuscript by Nov 04 2019 11:59PM. To enhance the reproducibility of your results, we recommend that if applicable you deposit your laboratory protocols in protocols.io, where a protocol can be assigned its own identifier (DOI) such that it can be cited independently in the future. For instructions see: http://journals.plos.org/plosone/s/submission-guidelines#loc-laboratory-protocols

We look forward to receiving your revised manuscript.

Kind regards,

Gavin Buckingham

Academic Editor

PLOS ONE

Journal Requirements:

Financial support was provided by the Centre for Robotics and Neural Systems at Plymouth University and by the Bavarian State Ministry for Science, Research & the Arts.

---

## [Author Response · Author response to Decision Letter 0]

27 Nov 2019

Response to the Reviewers

Reviewer #1

The authors study the link between spatio-temporal properties of lead-in movements prior to reaching in opposite force fields to address generalisation. It was found that changes in amplitude and speed of visual lead-ins had little effect on reach adaptation, whereas adaptation was affected when lead-ins were passive movements (involving somatosensory feedback) with similar variations in amplitude and speed. Overall the study is timely and interesting. The data is of good quality and the technical aspects of the research are sound. I do have several concerns and comments below, which I hope can strengthen the manuscript.

We thank the reviewer for their helpful comments on the manuscript. We address each concern in turn and indicate how we have updated the manuscript.

My first concern is linked to the theoretical framework used for interpretation. I follow the logic but I feel that the data was bent a little too much to fit in the generalisation framework, and I am not convinced that it is the best way to look at it. The main problem is that there is no statistical relationship between the lead-in movements and the force fields beyond the binary left-right info about direction. Thus, generalisation may not apply. It is perhaps more accurate to speak in terms of arbitrary association between lead-in directions and force fields, but the speed and extent of lead-ins were in fact irrelevant. Taking this into consideration the surprising finding was that the proprioceptive system seemed to not be able to form the same binary left-right association as the visual system. It sounds like a nuance but it is an important one. I believe that the main result is that different reach representations induced by somatosensory feedback depend on extent and/or speed, but it is a stretch to cast it into a generalisation study since there is no knowledge acquired about the statistics of lead-ins (again beyond binary left-right) used to predict the properties of the force field.

As the reviewer mentions this is a “nuance” such that they “are not convinced that this is the best way to look at it.” We do not think that this is a critical point of the manuscript, the data are presented clearly and the reader is free to make their own judgement on these cases. However, if we must choose between the reviewers proposed binary association of lead-in direction with the force field and our generalization as the kinematics of this lead-in movement changes, then only our generalization interpretation is supported by the results. Specifically, the reviewer’s proposal would predict no change in the predicted force output over the entire range of lead-in kinematics. Indeed, the reviewer makes the point that our results are “surprising” given their proposed binary interpretation. It is also important to point out that this decrement in predicted force output as the kinematics were varied occurred both for the passive and visual lead-in movements, as supported by the statistical analyses that have now been added to the manuscript. That is, the results clearly show that changing the speed and distance of the lead-in movements from the training condition values does affect the recall of compensation in both conditions. So, the results demonstrate that these kinematics are not irrelevant. However, we appreciate the different point of view and now discuss this possibility in the discussion. 

A second main concern is related to the presentation of the transfer (from one modality to the other) and also to the occasional substitution with a reversed cursor motion. I still do not understand the part with reversed cursor motion and I would recommend to describe it in more detail.

We have now explained the reversed cursor condition in greater detail.

A final major concern is linked to the experimental design. In fact, it is possible that participants learned the task perfectly in the passive lead-in case since generalisation trials were always followed by a channel (lines 266-267). I do not think that it would be difficult to figure out implicitly that once passive lead-ins are not the usual ones (based on somatosensory feedback about joint angles and speed), then one can predict no force field and as a consequence express low adaptation index away from the training lead-in. In such case it would be the visual system that did not generalise. Pls discuss this possibility or better justify if the data allows to reject it. It is possible that a control experiment without a systematic association between lead-ins and channels be necessary.

If the reviewer stated correctly their point that the sensorimotor control system implicitly determines the difference between the current lead-in movement and the trained lead-in movement producing a decrease in the predicted force compensation on these trials that decreases stronger for the passive condition than the visual condition, then we agree completely as this is exactly what we are concluding from our study. We refer to this decrease in the recalled force production as the lead-in kinematics varies away from the trained movement as generalization. In this case the visual system is not as sensitive to the difference in the lead-in movements (as we explain in our discussion) leading to a broader effect (although these values still decrease). We see no difference between this and what we discuss in our paper. Note that this reduction in predictive force compensation can be seen even in the very first channel trial (Additional Fig 1 & 2 below).

Specific comments:

Lines 33-34: “neural tuning of memory formation in state space”: this sentence make the abstract sound too focused and technical.

We have removed this statement in the abstract.

Lines 36-37: The link between kinematics of passive and visual lead-ins is very difficult to understand in the first pass and it is not clearly linked to the proposed gap in knowledge about how kinematics impact the representation of the next movement.

We have re-written this section. However, given the word limits for the abstract, further description in provided in the introduction where these aspects are described in more detail.

Abstract: overall difficult to follow at first.

We have revised the abstract. 

Intro: I felt that “lead-in” movement could be defined in more detail earlier on. 

As suggested, we now explain what we mean by a two-part movement with lead-in and probe movement in the abstract and in more detail in the introduction.

Fig. 3 the experimental blocks and the blocks of 8 trials used for display are not similar which is confusing.

The experimental block sizes were different in the various phases of the experiment to accommodate the training and generalization trials. Using a mean of 8 trials allows us to consistently use a fixed number of trials throughout the experiment. We now explain in the paper why we use 8 trials. 

Lines 275: was had

Corrected.

Line 310: Surely there is feedback during movement even in a channel, so the force measured is not a pure “feedforward” adaptation.

We have now expanded and re-rewritten the section of force estimation in more detail and by explain more on how the channel techniques operates and how channel trials do provide an estimate of feedforward adaptation. The critical point is that the force measured cannot be as a feedback response to the force field on these specific trials as there is no force field on these trials. This does not mean that all feedback is eliminated, but certainly its error-driven component is minimized.

Lines 392-393: The suggested lack of sensitivity of visual info to changes in lead-in kinematics is potentially simply do to the fact that participants only used binary info about cursor motion.

As responded above, we have added a discussion of this possibility to the discussion section and outlined why we believe that this is unlikely. It is important to note that even in the visual condition there is a decrease in the force output as the kinematics are varied, as presented in the results section of the manuscript. 

Lines 404-405: What was the rationale for using frequentist and Bayesian statistics?

We now only report frequentist statistics.

Lines 539-540: Typically an observer is used to describe the reconstruction of the state from partially observed systems. The Smith Predictor predicts the output of the system but it is not an observer in the sense that the state variables are not reconstructed. Pls check terminology or clarify.

We have revised this paragraph.

Line 553: The reconstruction of the state from prediction and feedback is the operation carried out by the observer. Pls clarify: the reconstruction of the state and the compensation for the delay are different operations.

We have revised this paragraph.

Reviewer #2

In this manuscript, "Asymmetry in kinematic generalization between visual and passive lead-in movements are consistent with a forward model in the sensorimotor system", the authors probe the generalization of lead-in movement kinematics as contextual cues for the simultaneous adaptation to opposing force-fields (an interference paradigm). Specifically, the authors have attempted to systematically probe the ranges of several kinematic parameters of lead-in movements (e.g., movement length, duration and peak speed) to assess to what degree this information can facilitate the adapted behavior. The study investigated the generalization and individual characteristics for both passive and visual lead-in movements. The results show that the force-field adaptation is sensitive to the duration and peek speed of lead-in movements but not to the distance, and the generalization between passive and visual lead-in movement was asymmetric due to different levels of sensory state estimation. While I find the study question and the results somewhat interesting, there are several critical issues regarding the statistical and the experimental design that make it difficult to draw meaningful conclusions from the results and overall lowers enthusiasm. Furthermore, there is the issue of a prior presentation/publication of aspects of this presented work that is not referenced in the current manuscript and appears not consistent with the presented results. My specific comments are shown below:

We thank the reviewer for their helpful comments on the manuscript. We address each concern in turn and also indicate how we have updated the manuscript.

Major Comments

1. I am confused as to why the conference paper from the 2018 International Conference on NeuroRehabilitation, Characterization of Neural Tuning: Visual Lead-in Movements Generalize in Speed and Distance by the authors, is not referenced in the manuscript. Much of the phrasing in the conference paper matches the current manuscript. Indeed, the subject information of the conference paper perfectly matches the subject information of the visual lead-in subjects in the current manuscript. Furthermore, while the visual lead-in experiment appears identical, a discrepancy exists between the percentage of force compensation seen in the subjects of the conference paper and the current manuscript. The conference paper states that subjects compensated up to 90% whereas subjects in the current manuscript compensated up to 70%. Also, Figure 1 of the conference paper appears identical to Figure 5D-F of the current manuscript, albeit with the force compensation axis shifted by 20%, the same difference is seen in the reports of force compensation percentage. In the conference paper, we see the location of the force compensation for the dotted line (represented by a black dot) at approximately 85%. In the current manuscript, we see the same shaped curves but this same location is at 70%. Clarification as to (1) why the conference paper was not cited/acknowledged (2) if the results are indeed from the same experiments or not, and (3) why the difference in force compensation percentage exists seems critical in order to assess this work.

A very early analysis of part of the data was presented in a short conference paper. We have since revised the analysis code, correcting several mistakes that were present in the early work and extending it, including a new experiment and comparisons across experiments. We now cite the conference paper as well.

2. Overall the description of the experiment is not clearly explained. The authors tried to investigate the kinematic generalization of velocity dependent force field adaptation on 34 generalization conditions, transfer between passive and visual lead-in movements, and transfer to a reversed visual condition in a very long experiment with over 3000 trials. In the manuscript, only the number of trials and type of trials were provided. Details about the structure of the experiments and how different trials are organized is currently missing. For example, how were catch trials distributed, what were the randomization and blocking rules? X per Y trials? Could catch trials be followed by a second catch trial? Did the rules change between pre-exposure, training, and testing? Without this information, it is impossible to understand such a complicated experiment and to verify whether the experiments are well designed to answer the questions. 

Furthermore, it is not clear to me that the experiments are able to sufficiently isolate the behavioral aspects that the authors set out to study, leading to a large uncertainty in the conclusions that can be drawn.

We have extensively revised the methods including all of these points.

3. Line 196-204: The Protocol for Experiment 1 states that lead-in movement direction was +/- 45 degrees from the final target direction. However, if I am interpreting it correctly, Figure 2 (labeled Experimental Design) shows lead-in movements +/- 135 degrees from the final target location. Thus, it appears that Line 197 and Line 202 are not consistent with figure 2 on the lead in movement directions. I assume that the figure was reused from a previous paper. Please replace with an updated figure showing the current experimental design. Furthermore, it is stated that final target locations are 0 and 270 degrees and lead-in movements begin from 45, 135, and 225 degrees. Given the final target locations, lead-in movements from 135 degrees could not be +/- 45 degrees from any final target location. Clarification and correction of the experimental design would be appreciated.

We thank the reviewer for pointing out the ambiguities in our description of the task here. Fig. 2 is actually correct and represents the 0� target location condition, but we appreciate that the description was confusing. Consequently, we have re-written this section to make it clearer.

4. Line 269-271: Experimental Design: How many trials were done with the reversed direction visual cursor? Additional explanation of the nature of this condition is necessary. 

The number of channel trials in each generalisation condition has now been documented in the text. We also provide an expanded description of the reversed cursor condition as suggested.

Nevertheless, I find the use of this type of contextual cue puzzling. If the reversed direction visual cursor begins from the center target and moves away from the hand position, how could this be considered a lead-in movement?

We do not claim that this is a lead-in movement. However, in the visual condition we had the opportunity to investigate the effect of the reversing cursor direction using a lead-away movement, which cannot have an equivalent passive condition version. We now describe this condition in more detail and refer to it as a “lead-away” movement.

5. Line 291-294: While both Bayesian and Frequentist ANOVA are valid approaches, it is not useful to apply both simultaneously. The assumptions about the nature of unknown parameters between Bayesian and Frequentist methods are in direct conflict. Thus, it appears that no new information can be garnered by using both approaches, only alternate interpretations.

We disagree with this statement. However, we now only report frequentist statistics for issues of space. 

6. Line 302: Maximum Perpendicular Error was calculated over 8 trials. It is unclear what this means (subset of trials used? moving average in reference to plotting?) Additionally, while MPE has been used in the past, it is not very informative regarding the nature of the error. It seems that choosing a controlled time or distance into the movement for kinematic error would be more informative.

We chose MPE to represent kinematic error to be consistent with both our previous studies, as well as those of many other researchers. It is important to note that the MPE values are not essential to the interpretation of the results which is made on the bases of the estimated recalled compensation to the viscous curl fields. We have also re-written this section to make it clearer. 

7. Line 314. To obtain the estimate of the level of force compensation, the measured force was regressed with the velocity of movement during the ‘same period’. No definition of the ‘same period’ was given. The authors should indicate how they aligned different movements in the ‘same period’ to calculate the force compensation. Also on Line 316 the authors state that the force compensation data was averaged across 2 channel trials for each participant. The authors should provide the definition of which 2 trials were used.

We now have re-written this section of the manuscript to explain much more clearly how compensation to adaptation was computed and why channel trials provide a good estimate of feedforward/predictive recall. We also explain that channel trials were not allowed to be adjacent or at first location in a block.

8. Line 344-346: As stated, Figure 3 shows force compensation averaged over both force fields. It would be beneficial to show compensation to each individual force field. Additionally, statistical testing seems warranted to check if asymptotic compensation levels are significantly different (or not) between both clockwise and counterclockwise force fields and between the two final target locations, as well as between day 1 and 2.

We have now added an additional plot to show compensation to each force field separately to the figure. There are no significant differences for either experiment.

9. Line 390-392: While I understand why the authors have chosen the wording used for the “one major finding” I fear that the phrase “overall tuning effects were much more pronounced” could be easily misinterpreted. The important results appear to be that visual lead-in movements, regardless of kinematics (to a degree), are able to act as contextual cues that result in behavior showing near trained condition levels; whereas the kinematics of passive lead-in movements restrict the degree to which lead-in movements can be utilized as a contextual cue. I feel this framing is critical for clear understanding for the implications of the results.

We disagree with the reviewer on this point. Distinct visual lead-ins associated with opposing curl fields certainly can act as a contextual cue, but recall is affected by changes in their kinematics. Similarly, distinct passive lead-ins associated with opposing curl fields can also act as a contextual cue, but recall is more sensitive to kinematics change. This is why we state that overall tuning effects were much more pronounced in the passive condition than in the visual condition.

10. Line 402-405: In the Factor Effects ANOVA, was a significant interaction effect found? Conclusion about the Main Factors cannot be made from the results of an ANOVA if a significant interaction exists, without additional testing. The authors should perform post-hoc tests which are currently missing in the manuscript. The authors should show the significance of the main effects (duration, velocity, distance, visual/passive, etc.) and the interaction between effects. Furthermore, the authors should find an appropriate multiple testing correction procedure to adjust the statistical confidence measures based on the number of tests performed. The authors might also consider the correlation between the data under different generalization conditions. Repeated measure ANOVAs or linear mixed models might be more appropriate for correlated data samples.

We calculate a repeated measures ANOVA to examine the difference in generalization between the visual and passive conditions. In order to do this comparison, all data is normalized to the level of the trained data – i.e. 100% for each subject. We then do the repeated measures ANOVA and find a significant between subject’s effect of the two experiments indicating that there is indeed a difference in the generalization of these two conditions.

11. Line 418-420: What statistical test was used in to find the significant difference between transfer from passive to visual vs. visual to passive? Furthermore, when presenting the results, the authors should list detailed information (specific numbers) about the results and perform appropriate statistical analysis before drawing any conclusion. For example, Line 370, “the recall of predictive compensation was strongly affected by changes in the duration of the movement”. Instead of saying “strongly affected”, the authors should show values of the recall and use hypothesis tests to show whether the recall for different movement durations is significantly different. Furthermore, the authors should provide detailed information about hypothesis tests that are used, main effects and interactions between effects they want to examine. The same issue holds for Line 363, 372, 377, 391, and 393. The authors should interpret the results by providing specific numbers supported by appropriate statistics.

We have included results of post-hoc comparisons in the results section for these details.

12. Is there a significant difference in transfer to Reverse Visual between subjects trained with passive lead-in movements vs. visual lead-in movements?

We have now added this test to the manuscript.

Minor Comments

1. Lines 37 and 86: The definition of the ‘lead-in movements’ should be given before use. In the current form it would be difficult for readers who lack the background knowledge to follow.

We now explain what we mean by a two-part movement with lead-in and probe movement in the abstract and in more detail in the introduction.

2. Are 8 subjects per group sufficient power for this study?’

Using 8 participants is a commonly used number of subjects for such behavioural motor control experimental studies.

3. Line 97: Small typo, “have been used effectively to examine the angular…”

Corrected.

4. Lines 225-229: The experiment is described to take place over two-day period. Were there any effects of having the experiment over these different days?

We ran the experiment over two days to collect more datapoints to reduce the variance of the compensation estimates used for the generalization conditions. This number of trials could not be run in a single session as it would be strenuous for the participants. We now explain this in the manuscript.

For the benefit of the reviewer, additional analyses were also carried out to show the effect of generalisation block count in the passive (Additional Fig. 1, below) and visual lead-in experiments (Additional Fig. 2, below). This shows that the effect of 2-day training was essentially to reduce the variance of the compensation estimates and does not affect the form of the generalisation functions or their interpretation. In addition, the same noisy trends are even present on the basis of the first generalisation condition block.

5. Line 226-227: While I assume that the large number of trials was motivated at least in part by the need to simultaneously train the different force-field conditions, a sentence explaining the motivation would be helpful for those who have less experience with interference paradigms.

We have added a statement explaining the motivation for a large number of trials.

6. Line 241-244: The number of each type of trial is included, but not the structure. How were the trials organized within each block?

We have extensively revised the methods section to include the details on the exact organization of trials. Briefly, the trials were all pseudorandomized within a block such that two channel (probe) trials never occurred directly one after the other.

7. Lines 379-387: There are multiple typos in this paragraph.

We have made corrections to this paragraph.

8. Lines 379-381: Small typo, “In the visual lead-in condition, we found slightly less pronounced decay as some of the kinematics were varied while others remained constant (Fig 5 D-F).”

Corrected.

9. Lines 382-383: Small typo, “Although there was a tuning effect centered around the movement duration using for training…”

Corrected.

10. Line 427-429: It would be helpful to state the actual value for transfer force compensation, not relative to the forward visual lead-in condition, and to also provide the standard error for all group means presented in the results section.

In the results section we have now added the actual values for transfer force compensation results in terms of the group means and standard error values.

11. Lines 431-433: What figure is the text referring to?

We now clarify this is referring to Fig. 6.

12. Line 460: “passive tuning was more pronounced and narrower…” This sentence seems misleading. Passive lead-in displayed lower asymptotic force compensation levels at the trained location than visual lead-in subjects. It seems all that can be said is that passive tuning was narrower.

Although this referred to previous studies, we have now deleted the statement “more pronounced” to make the statement less ambiguous.

13. When discussing the amount of force compensation during generalization condition trials, I would hesitate to call the reduced compensation “decay.” This might easily be conflated with having some temporal change over trials/time. “Reduction in force compensation” or phrasing along those lines may be more straightforward.

We have replaced the term decay with reduction where appropriate throughout the manuscript. In other places we have specified that this decay is spatial in nature rather than temporal. 

14. Line 500: Perhaps, “Further, experiments have shown that future state also affects motor learning in an analogous way”’

Was correct, but confusing. Have changed to “other experiments.” 

15. Lines 542-543: “This approach is also often used for engineering applications.

Corrected in a revised paragraph.

16. It would be helpful to add velocity information in table 1.

We have added the peak velocity information requested into Table 1.

17. As a supplement, a script of the verbal instructions given to subjects would be helpful.

We have added the instructions given to the participants to the methods section.

Additional Fig. 1 shows the effect of number of blocks used in estimation of the generalisation of passive movement lead-ins. Results in panels A-C show the estimates on the basis of all 16 blocks. Results in panels D - F show the analysis just on the basis of the first six blocks performed on the first day. Results shown in panels H - J show generalisation estimated simply on the basis of the very first block.

It can be seen as the number of generalisation blocks reduces from 16 to 6, the variance of the measurements of the estimates goes up, but the general trend is clear to see in all the plots. The single block estimates are quite noisy, but the overall trend is still visible. 

Additional Fig. 2 shows the corresponding effect that block number has on the visual-bead-in condition. Results in panels A-C show the estimates on the basis of all 16 blocks. Results in panels D - F show the analysis just on the basis of the first six blocks performed on the basis of the very first block. Again, it can be seen as the number of generalisation trials reduces, the variance of the measurements of the estimates goes up, but the general trend is clear to see in all the plots. Also, the single block estimates are again quite noisy, but the overall trend is still visible.

---

## [Decision Letter · Decision Letter 1]

18 Dec 2019

PONE-D-19-15362R1

Asymmetry in kinematic generalization between visual and passive lead-in movements are consistent with a forward model in the sensorimotor system

PLOS ONE

Dear Dr. Howard,

Thank you for submitting your manuscript to PLOS ONE. After careful consideration, we feel that it has merit but does not fully meet PLOS ONE’s publication criteria as it currently stands. Therefore, we invite you to submit a revised version of the manuscript that addresses the points raised during the review process.

Please pay particular attention to Reviewer 1's comment, which wasn't addressed in the manuscript and, in my opinion, somewhat weakly rebutted. Also Reviewer 2's comment regarding sample size needs some serious consideration - an n=8 per group may be standard in the field for large manipulations (i.e., prior research), but as the manipulations get more subtle and refined, we'd expect the power requirements to increase proportionally (particularly with so few trials per condition). Given the mandate of PLoS one to retain high levels of integrity over data and analytical methods, at the expense of considerations related to impact, this is something I feel particularly strongly about in this editorial role.

We would appreciate receiving your revised manuscript by Feb 01 2020 11:59PM. To enhance the reproducibility of your results, we recommend that if applicable you deposit your laboratory protocols in protocols.io, where a protocol can be assigned its own identifier (DOI) such that it can be cited independently in the future. For instructions see: http://journals.plos.org/plosone/s/submission-guidelines#loc-laboratory-protocols

We look forward to receiving your revised manuscript.

Kind regards,

Gavin Buckingham

Academic Editor

PLOS ONE

Reviewers' comments:

Reviewer's Responses to Questions

**Comments to the Author**

1. If the authors have adequately addressed your comments raised in a previous round of review and you feel that this manuscript is now acceptable for publication, you may indicate that here to bypass the “Comments to the Author” section, enter your conflict of interest statement in the “Confidential to Editor” section, and submit your "Accept" recommendation.

Reviewer #1: (No Response)

Reviewer #2: (No Response)

2. Is the manuscript technically sound, and do the data support the conclusions?

Reviewer #1: Yes

Reviewer #2: Yes

3. Has the statistical analysis been performed appropriately and rigorously? 

Reviewer #1: Yes

Reviewer #2: Yes

4. Have the authors made all data underlying the findings in their manuscript fully available?

Reviewer #1: Yes

Reviewer #2: Yes

5. Is the manuscript presented in an intelligible fashion and written in standard English?

Reviewer #1: Yes

Reviewer #2: Yes

6. Review Comments to the Author

Reviewer #1: The authors have addressed most of my concerns. I somewhat regret that they pushed back more than they attempted to verify their claim: one of my concerns was that should participants be able to learn that distinct kinematics are followed by channels, then perhaps the force applied to the channel wall decreased not because there was no generalization, but because participants knew there would be no force field. It is not an unreasonable objection, and it does not take anything away from the interesting finding that the visual system seems to operate differently. That some effect already happened in the beginning based on one block was not very convincing. I recommend that the authors either leave that option open and even discuss why they don’t think it could happen, or better, check it with a design in which generalization trials are not systematically followed by force channels.

Reviewer #2: The authors have addressed most of the previous comments. There are just a few points that still remain.

1) Of course there are previous studies that have used only 8 subjects to show reproducible effects, but the question is if 8 subjects is sufficient for the current study? At six trials per condition/subject (for 34 conditions) this is a reasonable question.

2) Did the authors actually mean 235 here and throughout the revision?:

“The 0o target location was associated with lead-in start locations at +135o and 235o, and the 270o target location was associated with lead-in start locations at +45o and 135o.”

3) Figure 2 could be clarified if the coordinate system was included with respect to the movements depicted.

4) For completeness it would be nice (but not critical) for the authors to show/state that the asymptotic training level of force compensation between passive and visual lead in groups in Figure 3 are not significantly different.

7. PLOS authors have the option to publish the peer review history of their article (what does this mean?). If published, this will include your full peer review and any attached files.

Reviewer #1: No

Reviewer #2: No

---

## [Author Response · Author response to Decision Letter 1]

27 Dec 2019

Response to the Reviewers

We thank both of the reviewers for their concerns and suggestions and we have addressed these issues in the current version. 

Reviewer #1: The authors have addressed most of my concerns. 

I somewhat regret that they pushed back more than they attempted to verify their claim: one of my concerns was that should participants be able to learn that distinct kinematics are followed by channels, then perhaps the force applied to the channel wall decreased not because there was no generalization, but because participants knew there would be no force field. It is not an unreasonable objection, and it does not take anything away from the interesting finding that the visual system seems to operate differently. That some effect already happened in the beginning based on one block was not very convincing. I recommend that the authors either leave that option open and even discuss why they don’t think it could happen, or better, check it with a design in which generalization trials are not systematically followed by force channels.

We disagree with the reviewer’s concerns here for several reasons. First of all, even experienced participants (which are not being tested here in our study, since participants were naïve) are normally unable to detect which trials are a channel trial and which contain a force field. Perhaps the reviewer may not have had experience with this set-up and experimental design which may lead to this confusion. Secondly, the reviewer states that they do not find the fact that we see our effects on the very first block of channel trials (a single channel trial for this condition) convincing. This is the most convincing result. The participants have no way of knowing before this point, that these test trials have a channel trial (instead of a curl field) following this slight variation in the preceding visual or passive motion. That is, there is no possible way for them to know a priori that there would not be a force field here. Even more important, participants can’t even detect after the trial has finished whether this was a channel trial or a force field trial, since the forces they experience in the two conditions are the same. The suggestion from the reviewer – either including a force field or having a null field – cannot be adopted to probe generalization as it would completely affect the results either by introducing learning or unlearning respectively. We have many more points to support our case, but perhaps the most relevant to this reviewer is that if this occurred as the reviewer suggests, then it would result in not the smooth variation in generalization forces on the channel but either force being produced or not being produced – i.e. a discrete rather than continuous effect on the generalization as the reviewer initially suggested and which we already discuss and outline why this is not the case within our discussion. We have added to the discussion outlining our position as requested. 

Reviewer #2: The authors have addressed most of the previous comments. There are just a few points that still remain.

1) Of course there are previous studies that have used only 8 subjects to show reproducible effects, but the question is if 8 subjects is sufficient for the current study? At six trials per condition/subject (for 34 conditions) this is a reasonable question.

There are several important points to make here. First of all, the reviewer is mistaken regarding the number of generalization trials. We have 16 repeated measurements for every channel trial condition. This means that each single point on our generalization measure is made up of the combination of 128 independent measurements. Moreover, our design does not require a single difference at any one of the various channel trials, but instead investigates the overall differences in the generalization function, whereby each generalization description is now made of 128 x 14 = 1792 trials. Here, eight participants provide an extremely strong and robust effect, which can easily be seen in our statistics. Particularly, all three comparisons of interest (visual generalization, passive generalization, and comparison of the two) produce main effect differences in our ANOVAs of p<0.001. Again, the critical point is that our results do not depend on any specific comparison of the 14 different conditions (although statistically speaking these are very strong as well), but explores the overall effects of varying kinematics. 

While perhaps a slightly orthogonal point, we believe that it is also important to point out that in our design that we do not simply record participants performing a few movements in 20 minutes of activity and then depend on statistics to detect a difference. Instead, by designing an experiment where we continue to measure participants behaviour over 6 hours of recording we are able to record sufficient data to see effects in a single participant. Our current manuscript is made up of 96 hours of individual movements from our participants. 

2) Did the authors actually mean 235 here and throughout the revision?:

“The 0o target location was associated with lead-in start locations at +135o and 235o, and the 270o target location was associated with lead-in start locations at +45o and 135o.”

We thank the reviewer for pointing this out. We have corrected it to 225�.

3) Figure 2 could be clarified if the coordinate system was included with respect to the movements depicted.

We have now indicated the movement directions in the legend of Figure 2.

4) For completeness it would be nice (but not critical) for the authors to show/state that the asymptotic training level of force compensation between passive and visual lead in groups in Figure 3 are not significantly different.

Although this comparison is not the point of our study, we now include this test. As expected from our figures, we found a significant difference between the asymptote of the two groups of participants using a t-test – and have added this result to the methods section.

---

## [Editor Report · Decision Letter 2]

8 Jan 2020

Asymmetry in kinematic generalization between visual and passive lead-in movements are consistent with a forward model in the sensorimotor system

PONE-D-19-15362R2

Dear Dr. Howard,

We are pleased to inform you that your manuscript has been judged scientifically suitable for publication and will be formally accepted for publication once it complies with all outstanding technical requirements.

With kind regards,

Gavin Buckingham

Academic Editor

PLOS ONE
---

## [Editor Report · Acceptance letter]

13 Jan 2020

PONE-D-19-15362R2 

Asymmetry in kinematic generalization between visual and passive lead-in movements are consistent with a forward model in the sensorimotor system 

Dear Dr. Howard:

I am pleased to inform you that your manuscript has been deemed suitable for publication in PLOS ONE. Congratulations! Your manuscript is now with our production department. 

With kind regards,

on behalf of

Dr Gavin Buckingham 

Academic Editor

PLOS ONE